# Promises and pitfalls of using LLMs to identify actor stances in political discourse

**Viviane Walker**, **Mario Angst**¤*

University of Zürich, Zürich, Switzerland

¤ Current address: Dr. Mario Angst, Digital Society Initiative (DSI), Zürich, Switzerland
* mario.angst@dsi.uzh.ch

## Abstract

Empirical research in the social sciences is often interested in understanding actor stances; the positions that social actors take regarding normative statements in societal discourse. In automated text analysis applications, the classification task of stance detection remains challenging. Stance detection is especially difficult due to semantic challenges such as implicitness or missing context but also due to the general nature of the task. In this paper, we explore the potential of Large Language Models (LLMs) to enable stance detection in a generalized (non-domain, non-statement specific) form. Specifically, we test a variety of different general prompt chains for zero-shot stance classifications. Our evaluation data consists of textual data from a real-world empirical research project in the domain of sustainable urban transport. For 1710 German newspaper paragraphs, each containing an organizational entity, we annotated the stance of the entity toward one of five normative statements. A comparison of four publicly available LLMs show that they can achieve adequate performance. However, results heavily depend on the prompt chain method, LLM, and vary by statement. Our findings have implications for computational linguistics methodology and political discourse analysis, as they offer a deeper understanding of the strengths and weaknesses of LLMs in performing the complex semantic task of stance detection. We strongly emphasise the necessity of domain-specific evaluation data for evaluating LLMs, considering trade-offs between model complexity and performance, as well as honestly weighing drawbacks of LLM application against traditional, valid approaches, such as manually annotating representative text samples.

## Introduction

Many areas of social science research are concerned with the normative positions, or *stances*, that social actors take in societal discourse. From communication studies over sociology to political science, the analysis of discourse helps to understand a crucial part of the social fabric, which shapes and is shaped by humans as social

**Data availability statement:** A replication package is available at https://doi.org/10.5281/zenodo.14795490.

**Funding:** This research was supported by grants from the DIZH, Digitalisierungsinitiative der Zürcher Hochschulen. The funder DIZH had no role in study design, data collection and analysis, decision to publish, or preparation of the manuscript.

**Competing interests:** The authors have declared that no competing interests exist.

actors. The analysis of discourse itself has a rich history. Here we follow an understanding of discourse itself, which emphasizes that discourse consists of "what is said", the ideas it revolves around, and, crucially, "who said what to whom where and why" [1] [p. 21]. This implies that stances taken by actors are an essential component of discourse, positioning social actors both in relation to the ideas that are the content of discourse and other actors, thus shaping discourse itself.

When social scientists embark on actually analyzing discourse, they often turn to the analysis of text. If the social scientist is concerned with identifying stances of actors as part of their analysis of discourse, expressions of such stances may be found in textual artifacts of various forms, both persistent and ephemeral, from newspaper articles and manifestos, over blog posts and letters, to slogans painted onto walls. Textual artifacts capture normative positions of actors at a given point in time and, in the ideal case, enable the analyst to retrace and make sense of societal discourse over a time period.

Analyzing textual artifacts has enabled social scientists to analyze discourse for a very long time, and in doing so, they have often *classified stances of actors*, teasing out the normative positions of actors. While such classification used to happen exclusively manually, the mass-production of textual artifacts as machine-readable texts, accompanied by advances in computing and software tools available to researchers, has led to a modern emphasis on understanding some forms of text increasingly as text-as-data [2], to be analyzed at a larger scale through distant reading and analysis by computational means. There is still a lot to be said for the unique understanding close reading and manual annotation of text can generate [3], but in the following, we will be concerned with automated text analysis used to analyze discourse.

The main-streaming of computational approaches to analyzing texts for social science research via automated text analysis, enabled by natural language processing, has led to stance classification as a core task of discourse analysis tasks to be increasingly delegated to machine learning models as well.

Stance detection, as viewed from a natural language processing perspective, is the automated classification of attitudes found in a text towards a specific organisation, topic or person [4]. It differs from the related technique of sentiment analysis which focuses on identifying positive, negative or neutral tonality within a text [5].

Recognizing attitudes toward a topic expressed in a text may appear relatively straightforward for a human in most cases. For computers, however, it has been generally very challenging to classify stances, even with the assistance of ever-increasing computing power. The classification of stances is especially difficult in the context of sarcasm, word ambiguity, implicitness, and the interpretation of sentences lacking substantial contextual cues [6].

Recent research claims that the pre-existing trade-offs between fine-tuning language models and developing rule-based stance detection systems have grown obsolete with the superior quality of results produced by LLMs [7]. Since fine-tuning LLMs is delicate and expensive, so-called prompt engineering for zero-shot classifications with LLMs has been proposed as a promising method for leveraging the language understanding abilities of LLMs [8].

While the potential for LLMs for stance classification has thus been heralded as promising, there currently is a lack of concrete evidence on their performance, as well as factors influencing their performance, in test cases representative of real-world empirical social science applications. This is especially true for non-English contexts. In this article, we start to address this gap by providing both a software implementation and an evaluation in a real-world, non-English applied social science research context of using LLMs to classify stances. We ask: Can zero-shot stance classification with LLMs achieve adequate performance in an applied, empirical social science research setting?

## Literature review and related work

### Stance detection as an automated text analysis challenge

The research field of automated text analysis is facing challenges regarding stance detection. According to [6], the presence of humour and irony is particularly demanding to handle. Texts containing irony are designed to be funny or to hide the truth and they can be mocking in nature. As a result, it can be challenging to ascertain the intended stance expressed with irony or sarcasm. Another challenge is word ambiguity, which occurs when a word has more than one meaning for its identical writing. The meaning of the word in question depends then on the context. Implicitness and the interpretation of sentences lacking sufficient contextual information within text data are additional challenges within this field of study. Early stance detection approaches relied on rule-based approaches. These approaches comprised systems that classified stances according to manually curated rules. Rule-based approaches are still very efficient in terms of inference costs, but are limited in their performance [5,9,10]. In order to detect stances with higher performance than with hand-crafted rules, the ability of understanding nuanced natural language expressed in texts is required.

Beyond rule-based approaches, stance detection approaches based on supervised machine learning with features or fine-tuning a pre-trained Transformer such as BERT [5,9,11] come closer to retrieving nuanced expressed stances. Supervised machine learning involves training algorithms on labeled data to learn patterns and predictions in order to predict stances accurately on unseen data. On the other hand, fine-tuning a pre-trained model such as BERT, which stands for Bidirectional Encoder Representations from Transformers [12], leverages a model that has already been trained in a self-supervised manner on a large corpus of text while adjusting on the fine-tuning data for the stance detection task specifically. In 2019, a study by [13] obtained results for stance detection on tweets and texts containing the stance *favor, against, or none* sourced from the publicly available datasets SemEval-2016 [4] and MPCHI [14]. Their method outperformed previous approaches using fine-tuned models. This improvement was largely due to the ability of language models like BERT to process context and nuanced language better, which is crucial for accurate stance detection.

A significant limitation of traditional supervised learning approaches is their dependence on quality labelled data, which is often unavailable or prohibitively expensive, given the vast number of human languages and the complexity of real-world natural language processing (NLP) problems. Consequently, supervised machine learning is often held back by training data availability and quality [9].

In the last years, researchers have developed ever larger (in terms of parameters and training data used) language models, following a general theory that with increasing size their performance would continue to improve [15]. Consequently, larger models are now often called large language models (LLMs) and consist of many more parameters than the original BERT model with 110 million parameters [12]. The evolution of language models has not only entailed changes in size and performance, but also in the methodology employed for their application. Unlike fine-tuning smaller models like BERT and its variants, fine-tuning LLMs with low or limited resources is significantly less effective [16]. Consequently, alternative techniques such as partial fine-tuning have been utilised, such as LoRA [17].

With the advent of large language models, approaches to adapting them to specific tasks and domains have shifted as well. One of these approaches is prompt engineering, which is the process of formulating an optimal prompt function for the LLM to perform the task. One can differentiate between zero- and few-shot prompting: Zero-shot prompting - which is

realised in this study - involves giving the LLM a task without any prior examples, while in few-shot prompting, the model is provided with a few examples to guide its response generation [8].

### Large language models for stance detection

The existing literature on stance detection with LLMs is currently still limited. [7] examine the discrepancy in stance detection efficacy between results provided by a LLM made available through a commercial offering called ChatGPT, provided by the company OpenAI (zero-shot approach) and established fine-tuned stance detection models. The study's understanding of the stance detection task is to predict a stance for an input (text) towards a target. The data used to evaluate all models was the dataset SemEval-2016 [4] and P-Stance [18]. The results demonstrate that the zero-shot approach with ChatGPT outperforms previous fine-tuned stance detection models, such as BERT. The findings of [7] contrast with those of [13] and highlight the power of LLMs in comparison to fine-tuned models.

[19] did not directly compare LLMs with smaller models but rather evaluated three established techniques for utilising LLMs in NLP tasks, called zero-shot, few-shot, and chain-of-thoughts zero-shot prompting. [19] demonstrated in their paper that while the chain-of-thoughts zero-shot approach could not surpass a model fine-tuned on human labels, it was overall the best-performing experiment.

The application of LLMs to stance detection presents several challenges and problems inherent to any application of LLMs [20]. Chiefly, these relate to their size, currently often legally dubious origins of training data, environmental impacts, propagation of bias, slow inference, and sometimes inaccessibility of models and therefore reproducibility of output, especially in the case of proprietary models and APIs. Considering the application of LLMs to a specific task, such as analysis of political discourse covered in this article, does therefore need to weigh such problematic aspects of LLM applications vis-a-vis promises of increased performance. In turn, to make this consideration, practitioners thus need more evidence on the actual performance of LLMs for specific tasks, especially evaluations of LLM performance on real-world, domain-specific datasets, as we present in this article. As such, our study does not present a new ground-breaking algorithm, nor provide new evaluations against generalizable benchmarks. We provide a data point from a specific use case, based on incrementally building on very much existing resources.

## Methods

All evaluation metrics and regression results can be replicated using data and code stored in an open online repository at https://zenodo.org/records/14795491. The repository also contains the evaluation dataset.

### Classification task

For each input text, a stance of a detected actor in relation to a predefined statement is to be predicted with one of the following stance labels: *support, opposition, irrelevant*. Fig 1 illustrates the general classification task.

### Evaluation data

The evaluation data used in this study is not a classical, English-language machine learning benchmarking dataset tuned to evaluate specific aspects of model predictions. Instead, it is representative of typical data that applied researchers in non-English speaking countries may encounter in automated text classification tasks of media data.

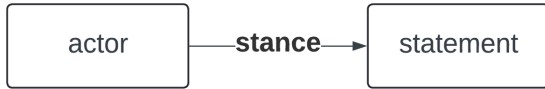

**Fig 1**. **Visualization of the classification task.**

The evaluation dataset in this paper consists of 1710 manually annotated German newspaper article paragraphs within the domain of sustainable urban transport discourse. More specifically, the dataset contains paragraphs on five specific discourse topics within this domain, which are *air traffic, parking spaces for motorized vehicles, driving speed and bicycle infrastructure*. For each of these topics a so-called policy core belief [21] was formulated in the form of a normative statement, capturing what we identified as the most salient point of debate within the discourse topic. These policy beliefs were inductively derived during a qualitative content analysis step within the larger research project the stance detection task was part of. The actual statements capturing policy core beliefs per topic are displayed in Table 1.

The articles, from which paragraphs were drawn, are sourced from online and print articles available through Swissdox@LiRI, a Swiss media corpus accessible to the authors. Articles originate from three major Swiss news publications (Tages-Anzeiger, Neue Zürcher Zeitung, 20 Minuten). In their political orientation, our media sources range from the more right-leaning Neue Zürcher Zeitung (NZZ) to the left-leaning Tages-Anzeiger [22]. The evaluation dataset covers paragraphs sampled randomly from a subset of paragraphs from all articles published in the newspapers containing 1) Zürich-related 2) sustainable transport topics, 3) with at least one organizational entity mentioned, based on a multi-stage analysis pipeline documented in [23]. Paragraphs span a time span of 13 years, from the beginning of 2010 until the end of year 2023.

Each paragraph in our evaluation dataset contains one or more organizational entities detected by a named entity recognition model. For every combination of entity and topic included in a paragraph, the stance of the entity toward the topic-specific policy core belief statement was annotated, as it could be inferred from the paragraph text, in the mutually exclusive three categories (see classification task above) support, opposition or irrelevant. The exact annotation task was defined in an annotation codebook, which explains each stance label and lists specific edge-cases (see Appendix section Codebook for manual annotation of stances by entities toward specific statements in newspaper paragraphs (translated from German)). An initial annotation batch (or batch 0, sampled in the same way as eventual gold-standard evaluation data) was utilised to get annotators acquainted with the data and to formulate the codebook. This data was afterwards discarded and not used for evaluation.

For actual annotation, the paragraphs were annotated individually by two to three annotators in four annotation batches. The first batch included 400 paragraphs and was annotated by two annotators, the second batch consisted of 443 paragraphs and was annotated by three annotators, the third batch contained 476 and was annotated by two annotators, and the last batch comprises 467 paragraphs annotated by two annotators, adding up to 1786 paragraphs in total. All annotation batches were annotated with a 100% overlap, meaning that every paragraphed was annotated separately by every annotator. In a subsequent step, disagreements were reviewed in a review round after completion of each batch in the whole annotation team to create the gold-standard labelled data, which was then used for evaluation. After all review rounds, excluding a small number of paragraphs for which no agreement could be reached, the final evaluation dataset consisted of 1710 paragraphs.

**Table 1. List of sustainable transport topics and associated policy core belief statements, toward which stances by actors were annotated in the evaluation dataset.**

| | Topic (English) | Policy core belief statement (German) | Policy core belief statement (English translation) |
|---|---|---|---|
| 1 | Air travel | Der Flugverkehr soll reduziert werden. | Air travel should be reduced |
| 2 | Parking spaces for motorized vehicles | Das Parkplatzangebot für den motorisierten Individualverkehr in der Stadt soll reduziert werden | The number of parking spaces for motorized individual traffic in the city should reduced. |
| 3 | Bicycle infrastructure | Das Fahrrad als Mobilitätsform soll gefördert werden. | The bicycle as a form of mobility should be promoted. |
| 4 | E-mobility | E-Mobilität in Form von E-Autos, E-Bussen, E-Scootern und E-Bikes soll gefördert werden. | E-mobility in the form of e-cars, e-buses, e-scooters and e-bikes should be promoted. |
| 5 | Driving speed | Zur Minderung von Emissionen soll die Fahrgeschwindigkeit in der Stadt reduziert werden. | Driving speed in the city should be reduced to reduce emissions. |

The annotation task proved to be hard for annotators, which sets an important baseline to consider when discussing the performance of the automated approach. The Inter-Annotator Agreement (IAA) of the first batch was 0.63, the subsequent batch with three annotators achieved an IAA of 0.41. The third and fourth annotation batches had an IAA score of 0.62 and 0.63 respectively. IAA values were calculated based on [24], with the R package caret [25]. These batchwise agreements result in a mean IAA over all batches of 0.55, and 0.62 for the batches annotated by two annotators, before the annotation review sessions. To re-iterate, the baseline against which we evaluate the LLM-based approach here is thus far from a stable "ground truth", however fraught this term is in social science context. Still, this is indicative of what applied researchers may face in many contexts. Additionally, we feel relatively confident in our evaluation data and that it represents the best quality we could achieve. This is due to the fact that beyond individual annotation, every annotation that made its way into the dataset either was independently annotated with matching annotations, or its label was discussed in the full annotation team until consensus was reached in case of initially disagreeing annotations.

## Evaluation setup

**Zero-shot prompting.** In this paper, we apply and evaluate zero-shot prompting with seven distinct prompt chains named *is, sis, nise is2, s2, s2is,* and *nis2e*, which we implemented in an open-source Python package called `stance-llm` (visit https://github.com/urban-sustainability-lab-zurich/stance-llm for more informations on the package). They are called prompt chains due to their multi-turn prompting, hence the LLMs stance detection procedure is done in an hierarchical manner. Depending on the prompt chain, the stance is classified in a different order. The specific prompt chains were formulated to explore four main concepts, in different combinations: 1) the inclusion of a summary step, 2) alternating orders of checks for relevance, 3) the inclusion of a dynamic, statement specific prompting step and 4) using constrained grammar, basing classification on generating text that is not a direct label.

The prompt chain *is* checks if there is a stance in the paragraph relevant to the statement or not. If the stance is relevant, the LLM is prompted to classify the stance as support or not support. If the stance is classified as not support, the final detected stance equals the stance label opposition.

*is'* progression is *sis*: The anterior *s* indicates the inclusion of summary prompt. For that the LLM is requested to summarise the paragraph, followed by the *is* prompt chain. The inclusion of a summary prompt was motivated by the hypothesis that a summary stage preceding the stance detection stage might mitigate known difficulties for stance detection tasks, especially related to convoluted sentence structure. The *nise* prompt chain is the most complex. It consists of a stance check that first checks if any stance is detectable in the paragraph. Then, if any stance was detected by the LLM, the LLM is prompted to classify, whether that stance is related to the statement, or not. If the stance is statement-related, the LLM is then asked to check in three separate steps, whether the stance is support, opposition or irrelevant.

The prompt chains *is2, s2is,* and *nis2e*, are special cases of the *is, sis* and *nise* prompt chains. The number two in a prompt chain label denotes the inclusion of a statement specific prompt in the prompt chain, where the statement is injected into a prompting step. This means that for the summary prompt, the LLM is not asked for a general summary, but for a summary in relation to the given statement, for which a stance is to be classified.

The prompt chain *s2* is a special case, because it directly prompts the LLM for a summary and stance in relation to the statement within one prompt. This leverages constrained grammar, prompting the models to start a text summary with a closed set of stance-specific signifiers.

Fig 2 displays a concrete example of the prompt chain *s2*. First, the LLM is prompted to create a summary of the detected actor's stance in relation to the statement. In a second step, the LLM is then instructed in a prompt to detect whether the detected actor holds a support, irrelevant or opposition stance within the summary text. The visualisations of the remaining prompt chains are viewable in Appendix section Prompt Chain Visualisations for all implemented chains.

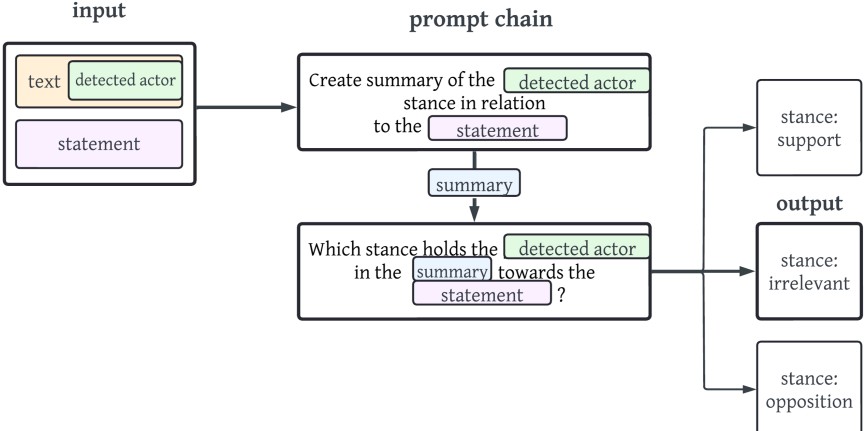

**Fig 2**. **Simplified and translated (exact prompts used are German) visualisation of the prompt chain *s2*.** The output of the stance irrelevant is shown with bold arrows and lines.

All prompt chains were implemented both with or without what we call *entity masking*. The masking of entities involved the replacement of the entity name within the paragraph text with a generic name, such as "Entity A". Controlling for entity bias is crucial to prevent LLMs from disproportionately emphasising specific stances linked to certain entities [26].

## Evaluated LLMs

We evaluated the performance of four different variants of the LLM Llama3, publicly released by the company Meta, for this study. To enable the replication of our results, and given data privacy and ethical concerns with relying on models provided by commercial vendors of LLMs, we relied only on openly accessible models. Since our evaluation data consists of German texts, we decided to only use specific German or multilingual Llama3 LLMs in our evaluation, which were fine-tuned on German data. We also decided to test three comparatively smaller 8B models (at the time of writing) with 8 billion parameters, which are likely more suitable for real-world social science applications with limited computing resources, and the largest possible model accessible to us at the time of writing, a 70B model with 70 billion parameters, which should in theory produce superior results. This led us to test the following four models, all freely accessible through repositories provided by the company Hugging Face: Llama-3-SauerkrautLM-70b-Instruct and Llama-3-SauerkrautLM-8b-Instruct by VAGOsolutions, Llama3-DiscoLeo-Instruct-8B-v0.1 by DiscoResearch, and Llama-3-KafkaLM-8B-v0.1 by seedboxai.

**Evaluation metrics.** Each German Llama3 variant was evaluated with each prompt chain on the entire evaluation dataset by calculating multi-class averaged precision, recall and F1 scores with the R package `yardstick` [27].

In the binary setting, thus having two stance classes only, precision and recall are calculated based on true positive (TP), false positive (FP) and false negative (FN) predictions.

$$\text{Precision}_{binary} = \frac{TP}{TP + FP} \tag{1}$$

$$\text{Recall}_{binary} = \frac{TP}{TP + FN} \tag{2}$$

The F1 score for the binary case is defined as:

$$F1_{binary} = \frac{2 \cdot \text{Precision}_{binary} \cdot \text{Recall}_{binary}}{\text{Precision}_{binary} + \text{Recall}_{binary}} \tag{3}$$

In our evaluation setting, note that we predict the class label *irrelevant* next to the labels *support* and *opposition*. With three labels, we thus need to average across classes. To do so, we use weighted macro averaging as implemented in `yardstick`, which calculates the metrics as binary for every class and then averages across classes, weighted by the relative true occurrence of the class in the evaluation set. For example, for the *F1 weighted macro score* this means:

$$\text{F1 weighted macro} = \sum_{i=1}^{N} \frac{n_i}{N} \cdot F1_{binary_i} \tag{4}$$

with $N$ denoting the total number of evaluation samples and $n_i$ denoting the number of evaluation samples per class label $i$. For weighted macro averaging of precision and recall measures, the procedure is equivalent.

**Evaluating influence of entity masking and statement.** The performance of our approach is likely to vary by the content of the statement toward which we detect stances. As our evaluation dataset contains five different statements, we are able to assess performance across evaluation runs depending on the statements.

To evaluate the dependence of the prediction quality on variance in statements, we use a logistic regression model of the form $y_i = \alpha + \beta_{statement}x_i$, where $y_i$ indicates the correctly predicted stance class (either true or false) for a given evaluation example $i$, $\alpha$ the intercept and $\beta_{statement}$ the slope of the categorical variable $x_i$, indicating the statement present in the evaluation example.

A similar approach can be taken to evaluate the influence of entity masking, although here, we evaluate by evaluation run and not by evaluation example. To evaluate the influence of applying entity masking across evaluation runs, we rely on a linear regression model of the form $y_i = \alpha + \beta_{masked}x_i$, where $y_i$ indicates the F1 weighted macro score for a given evaluation run $i$, $\alpha$ the intercept and $beta_{masked}$ the slope of the variable $x_i$ indicating the presence or absence of masking in an evaluation.

## Results

Evaluating the four distinct German LLMs, both with and without entity masking, resulted in a total of 56 sets of evaluation metrics (four LLMs evaluated on seven distinct prompt chains, with and without entity masking). In the following, we present the results of this evaluation matrix. First, we discuss the variation of metrics by LLM and prompt chain. Second, we explore variation in performance across different stance detection tasks present in the evaluation set, as our evaluation set contained five different statements to evaluate stances of entities toward. Third, we explore the influence of entity masking and statement on performance.

### Evaluation by LLM and prompt chain

Fig 3 displays F1 weighted macro scores across all LLMs and prompt chains with masked and non-masked entities colored grey and white respectively. The visualisations of precision and recall metrics can be found in the Appendix section Precision and recall values per LLM, prompt chain and entity masking.

The best performing combination in our evaluation matrix was the model Llama-3-Sauerkraut-70b-Instruct, the prompt chain *nis2e* and no entity masking, resulting in an F1 weighted macro score of 0.73. Across the statements, stances toward the statement "Air travel should be reduced" were evaluated with the highest F1 weighted macro score (0.87) with Llama-3-Sauerkraut-70b-Instruct. Most prompt chains achieved an F1 weighted macro score above 0.5 across all four LLMs, with the exception of the very badly performing prompt chain *s2*.

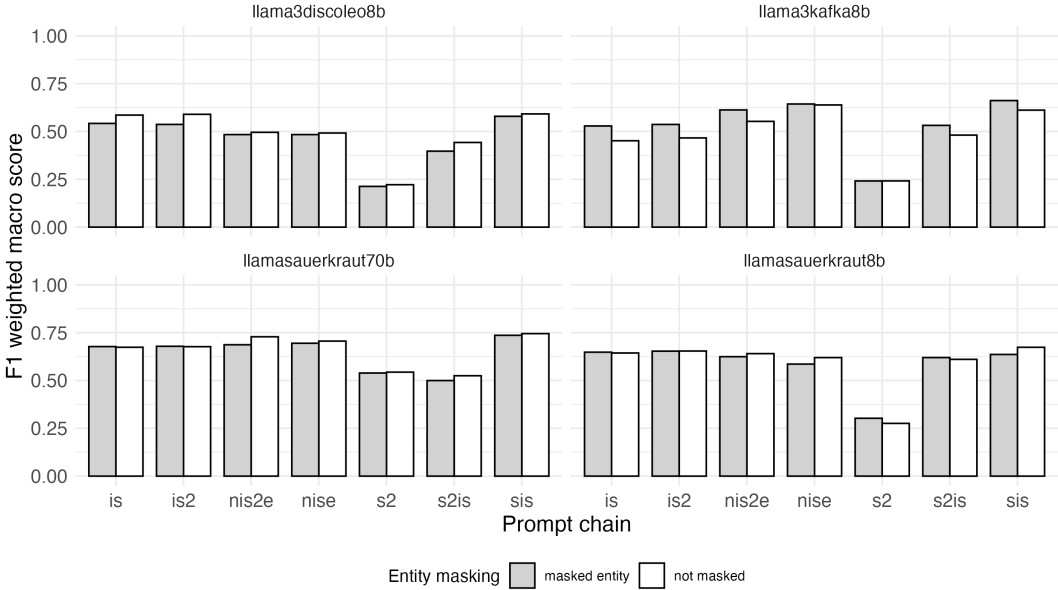

**Fig 3**. **Performance (F1 weighted macro scores) of zero-shot classification using large language models (LLMs) for stance detection task (prediction of stance of organizational entity regarding sustainable transport related normative statements in 1710 German newspaper paragraphs) by LLM, prompt chain and presence or absence of entity masking.** F1 weighted macro scores are calculated as the weighted average of the F1 binary scores for each stance class (support, opposition, irrelevant), where the weight for each class is proportional to its number of true positives.

### Evaluation by statement and influence of entity masking and statement

Fig 4 displays the distribution of F1 weighted macro scores by LLM for each of the five statements we evaluated (see Appendix section Precision and recall values per LLM, prompt chain and across statements for precision and recall weighted macro). We observe some variation in performance of our approach, depending on the statements, the LLMs were used to predict stances of entities toward. Generally, the performance of our approach is relatively stable across four out of five statement, but evaluations for one specific statement (on stances toward reduction of air traffic) score consistently higher across all LLMs evaluated.

The median F1 weighted macro scores for predicted stances toward reduction in air travel was 0.78, compared to median scores across all evaluated LLMs of 0.58, 0.57, 0.56, 0.55, respectively for the statements on parking spaces reduction, support of bicycle infrastructure, reduction of speed, and support of e-mobility.

Results of per-example regression modelling in Table 2 indicate that the null hypothesis of non-variance of correctly predicted stances across statements evaluated can be rejected at a significance level of $p<0.01$. Given the absence of a causal model, we can however not interpret the specific effects of statements further.

With regard to the influence of entity masking on performance, results in Table 3 of our per-run regression modelling indicate that the null hypothesis of entity masking not affecting performance cannot be rejected, given a significance cut-off of $p<0.1$.

### Error analysis

To uncover potential reasons for misclassifications done by the LLMs, we performed a qualitative error analysis on four distinct misclassification cases, as shown in Table 4. Case 1 encloses all cases where an LLM classified a stance as support instead of its true stance opposition. Case 2 looks at stances which where identified as opposition instead of

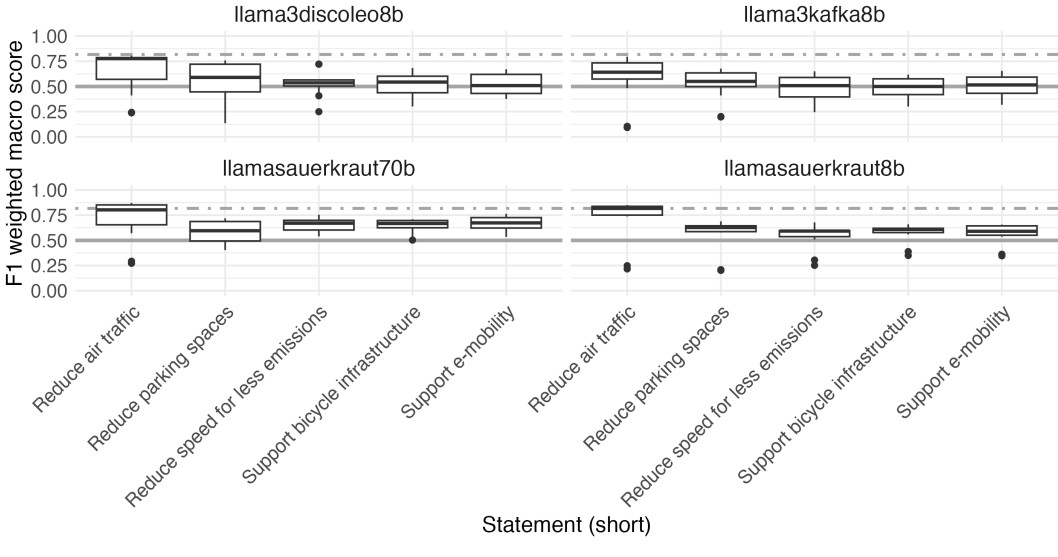

**Fig 4**. **Variance of zero-shot classification performance (F1 weighted macro scores) using large language models (LLMs) for stance detection task (prediction of stance of organizational entity regarding sustainable transport related normative statements in 1710 German newspaper paragraphs) by normative statement toward which stance was predicted.** The highest and lowest median are shown as reference lines in each plot. F1 weighted macro scores are calculated as the weighted average of the F1 binary scores for each stance class (support, opposition, irrelevant), where the weight for each class is proportional to its number of true positives.

**Table 2. Logistic regression results: Influence of statements on correct stance prediction per example.**

|  | Correct stance prediction |
|---|---|
| Reduce parking spaces | −0.409*** |
|  | (0.023) |
| Reduce speed for less emissions | −0.344*** |
|  | (0.026) |
| Support bicycle infrastructure | −0.303*** |
|  | (0.023) |
| Support e-mobility | −0.276*** |
|  | (0.023) |
| Constant | 0.591*** |
|  | (0.019) |
| Observations | 95,199 |
| Log Likelihood | -64,750.210 |
| Akaike Inf. Crit. | 129,510.400 |
| *Note:* | *p<0.1; **p<0.05; ***p<0.01 |

support stance. Case 3 focuses on the stances which where classified as irrelevant by the LLMs but in ruth where either support or opposition stances. Case 4 is the exact opposition of case 3: How many stances where misclassified as support or opposition instead of the true stance irrelevant? We focus particularly on these four cases since we think it is of importance to look deeper into cases where an LLM either classified exactly the opposite of the true stance (case 1 and case 2) or the misclassified the empirically prevalent classification of irrelevant (case 3 and case 4). A detailed, interactive and sortable table giving rates of all cases of error for every combination of our evaluation matrix can be found in the supporting materials and in the replication repository.

**Table 3. Linear model results: Influence of masking entities on the F1 weighted macro average score per run.**

|  | F1 weighted macro score |
| --- | --- |
| Entity masking | 0.015 |
|  | (0.038) |
| Constant | 0.535*** |
|  | (0.027) |
| Observations | 56 |
| $R^2$ | 0.003 |
| Adjusted $R^2$ | -0.016 |
| *Note:* | *p<0.1; **p<0.05; ***p<0.01 |

**Table 4. Error analysis of misclassifications: four cases.**

|  |  | true stance | | |
| --- | --- | --- | --- | --- |
|  |  | support | opposition | irrelevant |
| LLM prediction | support |  | case 1 | case 4 |
|  | opposition | case 2 |  |  |
|  | irrelevant | case 3 |  |  |

The detailed evaluation matrix reveals several critical patterns in stance detection errors across different cases. In case 1, where support stances were misclassified as opposition, the five highest misclassification rates all exceed 10 %, peaking at 14.1 %. Notably, the 8B-parameter LLMs (llamasauerkraut8b and llama3kafka8b) consistently struggled with classifying stances toward reducing parking spaces, particularly when classified through the (although generally badly performing) s2 prompt chain. Similarly, in case 2, where opposition stances were misclassified as support, the same statement about reducing parking spaces emerged as a recurring challenge, alongside misclassifications related to e-mobility. While the maximum error rates in this case were generally lower than in case 1, the sis and s2 prompt chains appeared to exacerbate these errors. As models seem to struggle specifically with identifying non-irrelevant stances toward reducing parking space, this needs to be put into context of the statement being very hard to annotate for human annotators as well. The IAA of for the annotation of stances regarding the reduction of parking space was quite low (0.4) before group review.

In case 3, where irrelevant stances were misclassified as either support or opposition, the models llama3discoleo8b and llamasauerkraut8b misclassified one-third of all stances, with pronounced errors occurring for the e-mobility support statement across multiple prompt chains (sis, is, is2, and nise). This suggests a broader vulnerability in the models' ability to accurately identify irrelevant positions on this topic. Finally, case 4, which encompasses support or opposition stances misclassified as irrelevant, exhibits the highest error rates overall, with approximately one-fifth of all evaluated instances exceeding 50 % misclassification. All four LLMs demonstrated particular difficulty with the air traffic reduction statement when paired with the s2 prompt chain. Across all cases, entity masking did not exhibit any consistent pattern in influencing misclassification rates, indicating that its introduction does not introduce any consistent error pattern deviating from not masking entities.

## Discussion

Our results give insights on leveraging LLMs for stance detection in applied social science research. On the one hand, results speak to the potential of LLMs for stance detection and hint toward best practices in applications. On the other hand, results show potential limitations and biases with zero-shot prompting specifically.

First, we evaluated a number of LLMs, which crucially differed in size between 8B (8 billion parameters) and 70B (70 billion parameters) models. The much larger 70B model outperformed the smaller models, which is unsurprising. However, what is interesting is the magnitude of the difference in performance, keeping in mind that at upper ends of

performance level for machine learning models in general, performance gains are usually not linearly scaling with model complexity. 70B models show increased performance across all implemented prompt chains, but some 8B models almost approached their performance for the best performing prompt chains. Given that larger models come with much higher inference costs and are much more challenging to deploy, this suggests that smaller models may be worth applying in applied research contexts, if a well performing application method can be found.

Second, we evaluated a number of differently constructed prompt chains. Results show that of the two most performant prompt chains (*sis* shown in Fig 6, and *is2* displayed in Fig 8, in our evaluation), one is relatively short in comparison and both contain a summary step. In a summary step, we prompted the LLM to summarize the input text with regard to the statement to evaluate against first, before classification. Going forward, these results suggest that a lower number of components within a prompt chain, if feasible, is preferable and that complex classification tasks can be improved by leveraging summarization capabilities of models.

Third, when it comes to limitations and biases to be aware of in applying LLMs for stance detection, entity masking (thus masking the entity for which the stance is evaluated by substitution with a generic string) should be considered. In our evaluations, entity masking did not lead to statistically detectable differences in performance across evaluation runs. These results should not be considered evidence for absence of bias, as we tested for impacts on performance, not bias. Rather, taking a precautionary approach, given that evidence strongly suggests that applications of LLMs always need to consider subtle and not so subtle biases [28,29], our results support the application of entity masking, given that performance penalties seem minimal. Further, our application within a relatively narrow, local policy context was not especially politically sensitive. However, when applying LLMs for stance detection, it should be considered that the debate on "political leanings" of model outputs is very much unresolved [30], again emphasizing the role of domain-specific test sets, especially for situations where classifications of stances might happen in sensitive domains.

Fourth, our results show that performance depends to an extent on specific statements stances are evaluated against. In our case, zero-shot prompting performed particularly well in relation to predicting stances toward reducing air traffic, and to a lesser extent, reducing parking spaces and lowering speed limits to decrease emissions. This variability may be attributed to the inherent differences in stances on these issues or to the linguistic complexities involved in detecting them. A similar pattern was observed during the manual annotation process, though it was most pronounced in the case of parking space reduction, which exhibited relatively low agreement among human annotators. Overall, using LLMs for stance detection is thus very likely not statement-agnostic and researchers should be careful with assuming that evaluation results in different contexts may apply to their specific research domains.

Lastly, we want to put our results in the context of the overall challenge of annotating real-world data. The annotation of stances on real-world data presents a significant challenge not only for statistical models, but also for human annotators, as qualitative researchers have known for decades (and as the IAA scores reported in this article show). The performance of automated stance detection approaches therefore also should not be judged against unrealistic expectations. Properly annotating or classifying stances is hard because ambiguities and nuances in language, such as sarcasm or irony, can make it challenging to accurately capture stances, as mediated through a written medium, but also because of the real-world complexity of stances held by humans, which might not always neatly confirm to the categories we want to classify them into.

## Limitations

Our study faces a number of limitations, which we see mostly as indicative of the need for more research to paint a more complete picture with regard to where our results have limited generalizability.

First, and indeed, generalizability is the most important limitation of our study. We are at the far opposite end of a specific view of generalizability in evaluating automated text classification approaches with our evaluation set, compared to how benchmarking of machine learning models is usually carried out. Crucially, our evaluation set is locally specific,

domain specific and non-English. As such, we obviously cannot claim to make statements that can translate directly to another location, domain or language. However, the insights we can make and have listed above do point toward starting points other applications can build on, in our opinion.

Second, another limitation of our analysis is that we do not carry out a detailed assessment of computational costs and potentials for reducing it. The approach described in this study would likely need substantial optimization to be used in applications where low latency is important or datasets are much larger.

Third, our analysis evaluates against human annotation, but not against other, more traditional approaches to stance detection, such as rules-based models. This was mostly due to relatively well established challenges to applications of such models in our specific context, but it is likely that such models might be much more amenable as a baseline or computationally much less intensive alternative in other, e.g. English-speaking contexts.

Fourth, another limitation of our study relates to the context supplied to annotators and LLM classification. We annotated stances within newspaper paragraphs, mostly due to considerations of computational efficiency in data processing (as the stance detection model was part of a larger processing pipeline) and to be able to create a test set of similarly sized, comparable units. However, the larger context of the overall article a paragraph is part of is lost in this approach and likely could have improved performance for human annotators as well as for the statistical models.

## Conclusion

Many months into what can only be described as a hype around LLMs, many researchers across different fields of the social sciences are finding out that their application is no panacea for substantially complex research problems. Rather, utilising these general-purpose language models to specific tasks in empirical research brings with it new challenges, alongside marginal benefits. Our analysis of employing and evaluating LLMs in an applied social science research context for stance detection is very much in line with this.

Stance detection is a complex process, both for humans and statistical models. In this study, we evaluated one-shot stance classifications with LLMs towards five distinct statements. Our results warrant at best cautious optimism regarding the application of LLMs for stance detection in applied empirical social science contexts. Given evaluation metrics obtained, applied researchers may experiment with, but not blindly utilize them. In many ways, this is the overall conclusion of our study: LLMs are general purpose models, which need high quality, context-sensitive evaluations, if applied in specific contexts. The drawbacks of applying LLMs as part of an analysis pipeline should always be honestly weighed against existing approaches with proven ways to ensure validity. As in many areas of social science methodology, progress requires translation of theoretical models into application and continuous testing and re-testing of tools, gathering evidence on performance and bias across diverse application contexts. We look forward to seeing more studies in this vein.

## Appendix

**Codebook for manual annotation of stances by entities toward specific statements in newspaper paragraphs (translated from German)**

**Overall goal of annotation**

What is the stance of the actor toward the belief statement?

### Three stance categories

- Support: the actor supports the belief statement explicitly or on a general level
- Against: the actor does oppose the belief statement
- Irrelevant: Paragraph does not contain any stance of the actor, although the actor might have a connection to the belief statement.

### Implicit versus explicit stances

Implicit stances are hard to annotate, especially judging the level at which something becomes too implicit. Examples for degrees of implicitness considered too implicit:

- In arguments to change the types of parking spaces (e.g. resident-only vs. free), the stance of the actor should be labeled irrelevant if their argument rests on e.g. protecting residents, implying no stance in support or opposition of parking spaces in general, but a tangential debate
- actors arguing for a "car-free" city do not necessarily imply support a reduction of parking slots (e.g. some people could think that reducing them increases traffic for search)
- also, "car-free" may occur in association with an argumentation for producing more pressure on the resident-only parking in the near neighbourhood

### List of specific edge cases encountered during annotation:

- If actor has two different stances from two different time periods, annotate most recent stance
- The electrification and digitalisation of public transport (ÖV) is considered as a support stance, specifically in the context of the statement "support e-mobility"
- annotate paragraph text with stance class support, if the actor and their product occurs together without an explicit stance since their product can signify a support of that belief statement, e.g. "Tesla" and "Elektroauto", "Voi" and "E-Trotti", "Publibike stellt E-Bikes zur Verfügung".
- expressions like "In Grenzen halten/Im vernünftigen Rahmen" in relation to the belief statement mean, that something, e.g. emissions, should not increase (just as the belief statement of emissions is defined). In other words: "... müsste reduziert werden, wenn Rahmen überschritten würde", is understood as a support of that reduction in case of an increase, hence a support stance.
- an action of an actor described in the paragraph can also be a considered as an expression of a stance
- if an actor is against or supports a temporary or compensatory solution, e.g. Parkplatzkompromiss, label the stance irrelevant

**Prompt Chain Visualisations for all implemented chains**

**Prompt chain is**

Fig 5 displays the concrete example of the prompt chain *is*.

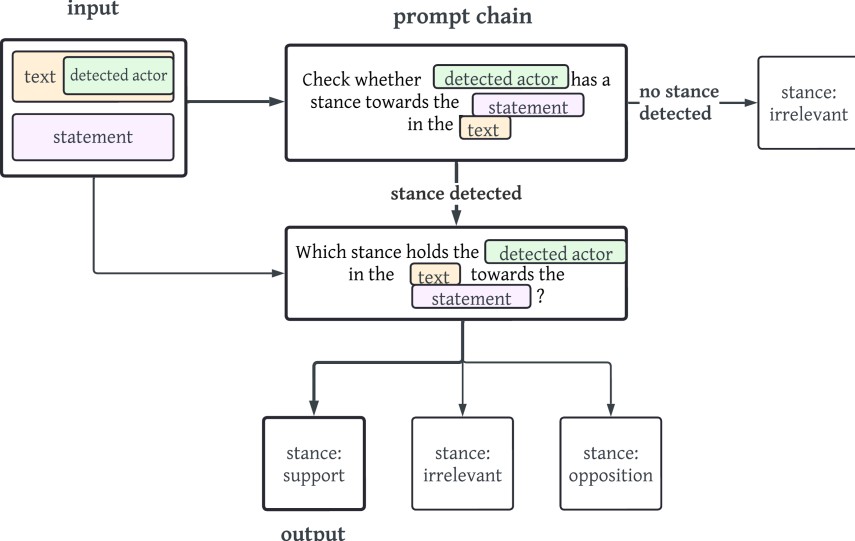

**Fig 5. Simplified visualisation of the prompt chain *is*.** The output of the stance support is shown with bold arrows and lines.

## Prompt chain sis

Fig 6 displays the concrete example of the prompt chain *sis*.

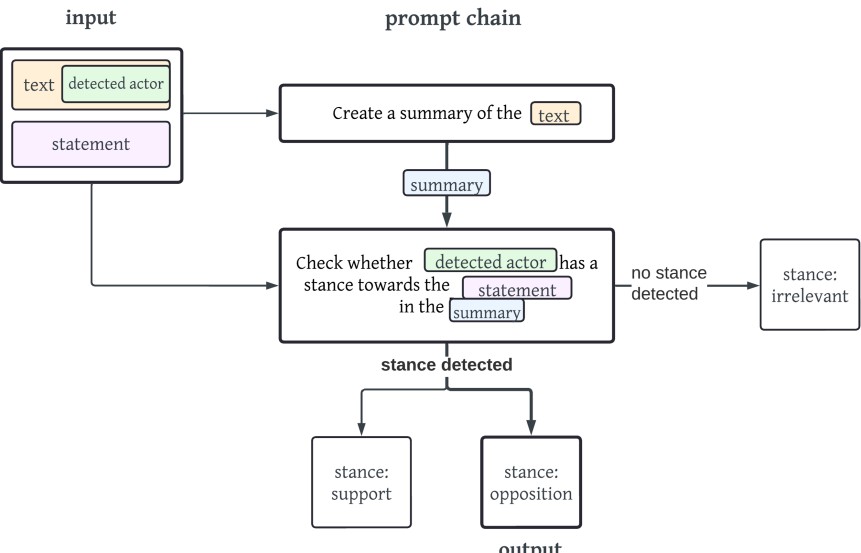

**Fig 6. Simplified visualisation of the prompt chain *sis*.** The output of the stance opposition is shown with bold arrows and lines.

## Prompt chain nise

Fig 7 displays the concrete example of the prompt chain *nise*.

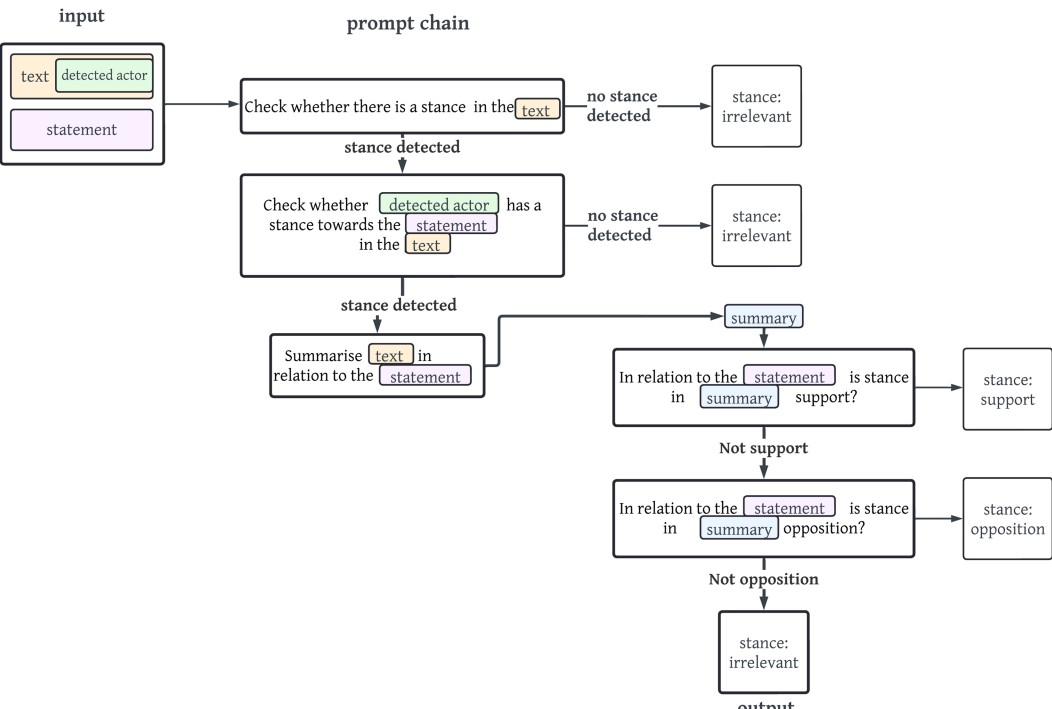

**Fig 7**. **Simplified visualisation of the prompt chain *nise*.** The output of the stance opposition is shown with bold arrows and lines.

## Prompt chain is2

Fig 8 displays the concrete example of the prompt chain *is2*.

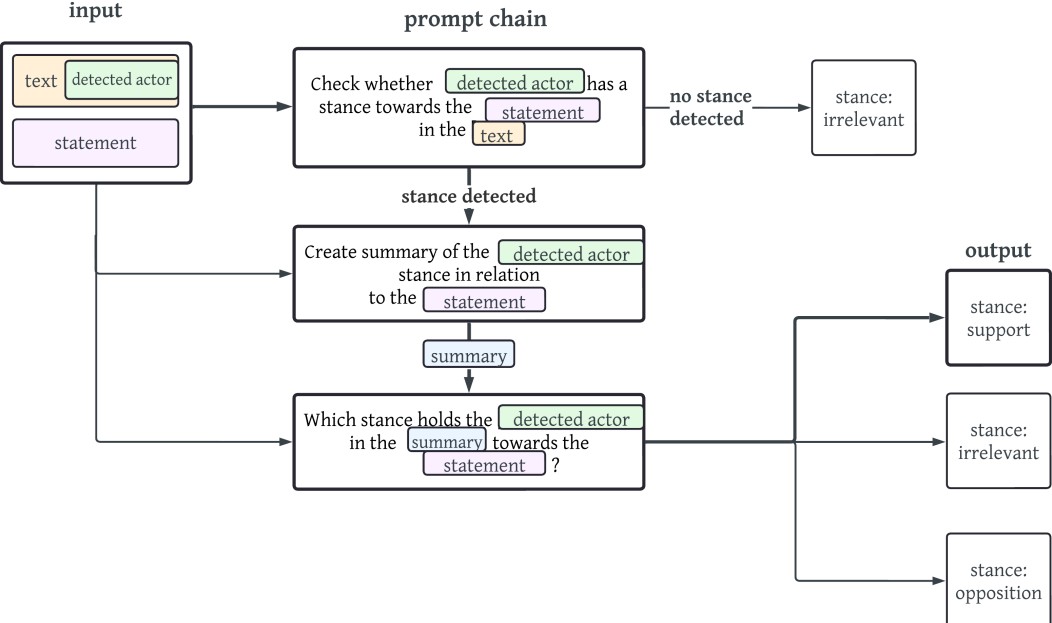

**Fig 8. Simplified visualisation of the prompt chain *is2*.** The output of the stance support is shown with bold arrows and lines.

## Prompt chain s2is

Fig 9 displays the concrete example of the prompt chain *s2is*.

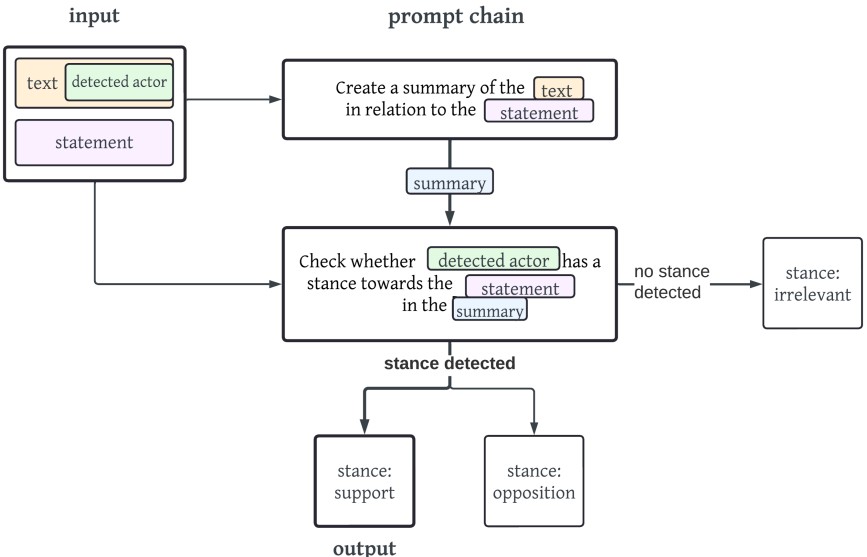

**Fig 9. Simplified visualisation of the prompt chain *s2is*.** The output of the stance support is shown with bold arrows and lines.

## Prompt chain nis2e

Fig 10 displays the concrete example of the prompt chain *nis2e*.

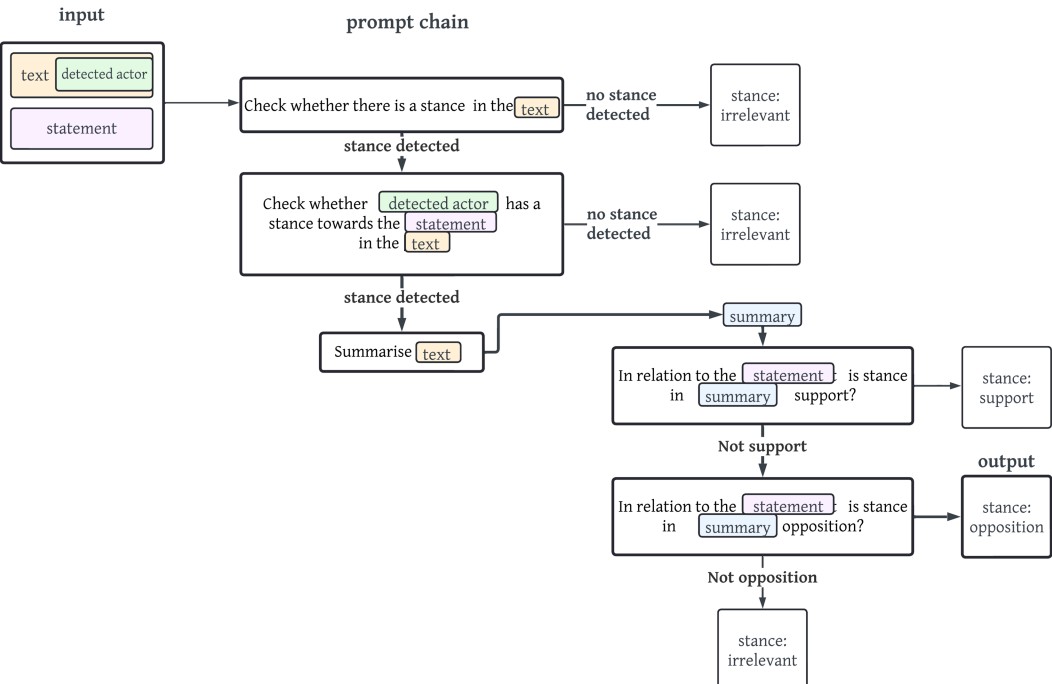

**Fig 10**. **Simplified visualisation of the prompt chain *nis2e*.** The output of the stance irrelevant is shown with bold arrows and lines.

## Precision and recall values per LLM, prompt chain and entity masking

Fig 11 displays the distribution of the metric precision weighted macro by LLM for each of the seven prompt chains we evaluated, as well as the absence and presence of entity masking.

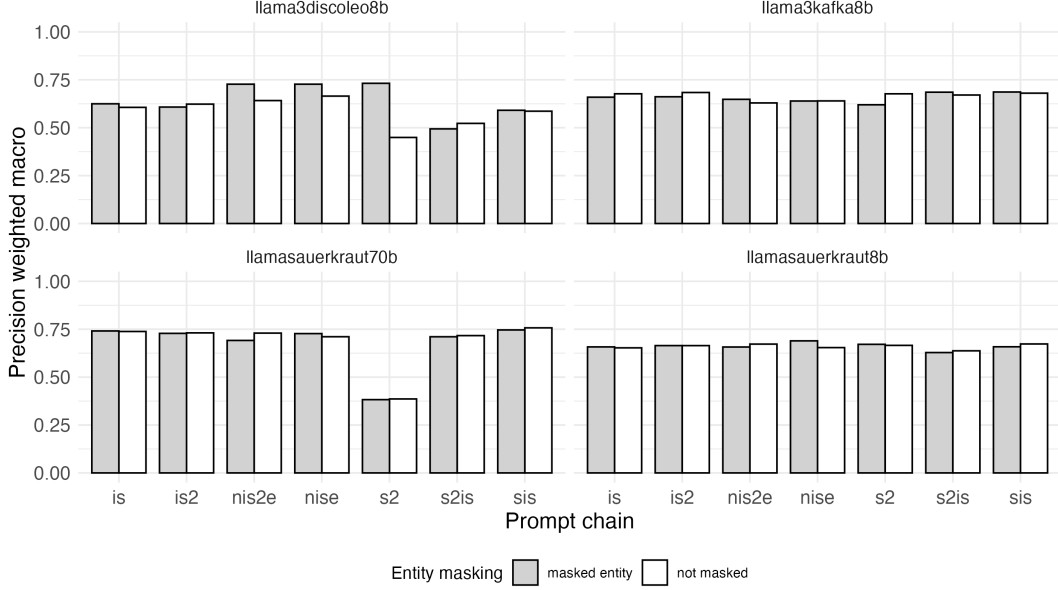

**Fig 11**. **Precision (weighted macro averaged) values of zero-shot classification using large language models (LLMs) for stance detection task (prediction of stance of organizational entity regarding sustainable transport related normative statements in 1710 German newspaper paragraphs) by LLM, prompt chain and presence or absence of entity masking.**

Fig 12 displays the distribution of the metric recall weighted macro by LLM for each of the seven prompt chains we evaluated, as well as the absence and presence of entity masking.

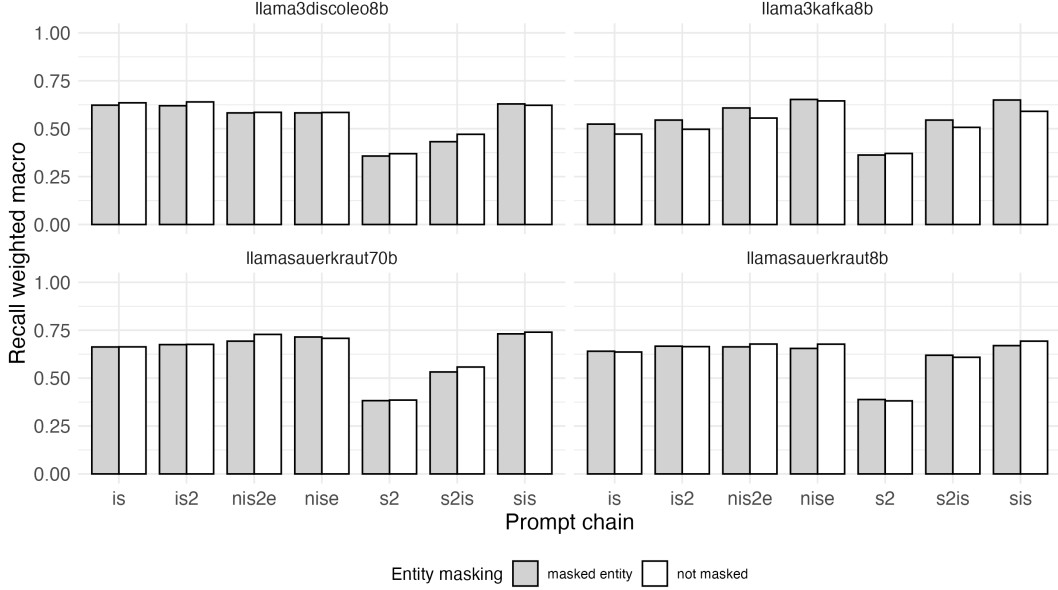

**Fig 12. Recall (weighted macro averaged) values of zero-shot classification using large language models (LLMs) for stance detection task (prediction of stance of organizational entity regarding sustainable transport related normative statements in 1710 German newspaper paragraphs) by LLM, prompt chain and presence or absence of entity masking.**

## Precision and recall values per LLM, prompt chain and across statements

Fig 13 shows the distribution of the metric precision weighted macro by LLM for each of the five statements we evaluated.

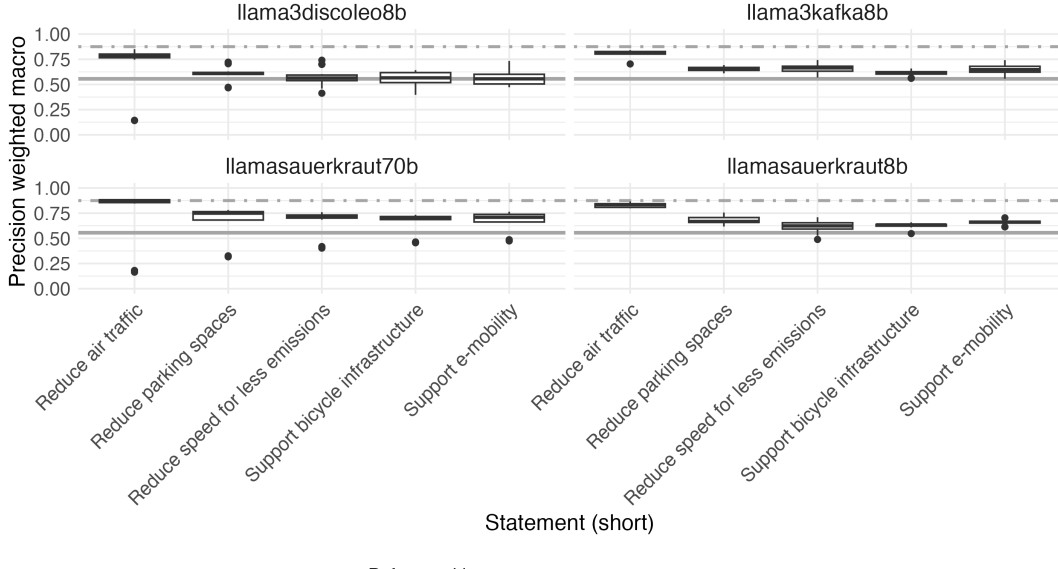

**Fig 13. Precision (weighted macro averaged) values of zero-shot classification using large language models (LLMs) for stance detection task (prediction of stance of organizational entity regarding sustainable transport related normative statements in 1710 German newspaper paragraphs) by LLM, prompt chain and across different normative statements toward which stances were classified.**

Fig 14 displays the distribution of the metric recall weighted macro by LLM for each of the five statements we evaluated.

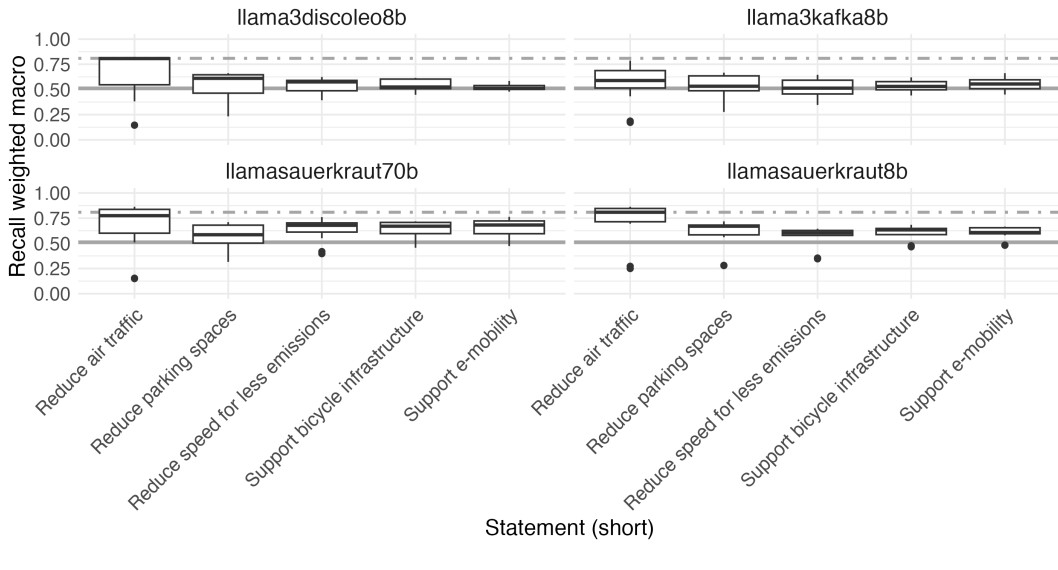

**Fig 14. Recall (weighted macro averaged) values of zero-shot classification using large language models (LLMs) for stance detection task (prediction of stance of organizational entity regarding sustainable transport related normative statements in 1710 German newspaper paragraphs) by LLM, prompt chain and across different normative statements toward which stances were classified.**

## Author contributions

**Conceptualization:** Viviane Walker, Mario Angst.

**Data curation:** Mario Angst.

**Formal analysis:** Viviane Walker, Mario Angst.

**Funding acquisition:** Mario Angst.

**Investigation:** Viviane Walker, Mario Angst.

**Methodology:** Viviane Walker, Mario Angst.

**Project administration:** Viviane Walker.

**Resources:** Viviane Walker, Mario Angst.

**Software:** Viviane Walker, Mario Angst.

**Supervision:** Mario Angst.

**Validation:** Viviane Walker, Mario Angst.

**Visualization:** Viviane Walker.

**Writing – original draft:** Viviane Walker, Mario Angst.

**Writing – review & editing:** Viviane Walker, Mario Angst.

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
