## [Decision Letter · Decision Letter 0]

2 Jun 2025

PONE-D-25-07214Promises and pitfalls of using LLMs to identify actor stances in political discoursePLOS ONE

Dear Dr. Angst,

Thank you for submitting your manuscript to PLOS ONE. After careful consideration, we feel that it has merit but does not fully meet PLOS ONE’s publication criteria as it currently stands. Therefore, we invite you to submit a revised version of the manuscript that addresses the points raised during the review process.

We look forward to receiving your revised manuscript.

Kind regards,

Thomas W. Sanchez, PhD

Academic Editor

PLOS ONE

**Journal Requirements:**

1. When submitting your revision, we need you to address these additional requirements. Please ensure that your manuscript meets PLOS ONE's style requirements, including those for file naming. The PLOS ONE style templates can be found at https://journals.plos.org/plosone/s/file?id=wjVg/PLOSOne_formatting_sample_main_body.pdf and https://journals.plos.org/plosone/s/file?id=ba62/PLOSOne_formatting_sample_title_authors_affiliations.pdf 2. Thank you for stating in your Funding Statement: This research was supported by grants from the DIZH, Digitalisierungsinitiative der Zürcher Hochschulen. The funder DIZH had no role in study design, data collection and analysis, decision to publish, or preparation of the manuscript. Please provide an amended statement that declares *all* the funding or sources of support (whether external or internal to your organization) received during this study, as detailed online in our guide for authors at http://journals.plos.org/plosone/s/submit-now. Please also include the statement “There was no additional external funding received for this study.” in your updated Funding Statement. Please include your amended Funding Statement within your cover letter. We will change the online submission form on your behalf. 3. Thank you for stating the following in the Acknowledgments Section of your manuscript: This research was supported by grants from the DIZH, Digitalisierungsinitiative der Zürcher Hochschulen. The funder DIZH had no role in study design, data collection and analysis, decision to publish, or preparation of the manuscript. For this publication, use was made of media data made available via Swissdox@LiRI by the Linguistic Research Infrastructure of the University of Zurich (see https://www.liri.uzh.ch/en/services/swissdox.html for more information). We note that you have provided funding information that is not currently declared in your Funding Statement. However, funding information should not appear in the Acknowledgments section or other areas of your manuscript. We will only publish funding information present in the Funding Statement section of the online submission form. Please remove any funding-related text from the manuscript and let us know how you would like to update your Funding Statement. Currently, your Funding Statement reads as follows: This research was supported by grants from the DIZH, Digitalisierungsinitiative der Zürcher Hochschulen. The funder DIZH had no role in study design, data collection and analysis, decision to publish, or preparation of the manuscript.  Please include your amended statements within your cover letter; we will change the online submission form on your behalf. 4. Please note that your Data Availability Statement is currently missing the repository name. If your manuscript is accepted for publication, you will be asked to provide these details on a very short timeline. We therefore suggest that you provide this information now, though we will not hold up the peer review process if you are unable.

**Additional Editor Comments:**

I think the three reviewer's have provided very useful comments and suggestions for the submission. I think the authors should read each of the comments carefully and respond to each in a revised version.

Reviewers' comments:

Reviewer's Responses to Questions

**Comments to the Author**

1. Is the manuscript technically sound, and do the data support the conclusions?

Reviewer #1: Yes

Reviewer #2: No

Reviewer #3: Yes

2. Has the statistical analysis been performed appropriately and rigorously?

Reviewer #1: Yes

Reviewer #2: No

Reviewer #3: Yes

3. Have the authors made all data underlying the findings in their manuscript fully available?

Reviewer #1: Yes

Reviewer #2: No

Reviewer #3: Yes

4. Is the manuscript presented in an intelligible fashion and written in standard English?

Reviewer #1: Yes

Reviewer #2: No

Reviewer #3: Yes

5. Review Comments to the Author

**Reviewer #1:** This manuscript examines the application of Large Language Models to stance detection in political discourse. The research is important and timely, as it systematically evaluates different LLMs for a specific NLP task with real-world applications. As more researchers consider incorporating LLMs into their analytical workflows, empirical assessments of their capabilities on domain-specific tasks provide valuable guidance.

The authors test LLMs on German newspaper content, which offers a more realistic assessment than using standard benchmark datasets. However, this approach introduces its own limitations regarding generalizability beyond German media discourse on transportation policy. The following sections reports my considerations and improvement areas.

Methodological Considerations

The evaluation approach tests various prompt chain structures, compares models of different sizes, and examines performance across statements. This multifaceted approach is methodically sound, though the number of tested variables creates challenges for isolating key factors driving performance differences.

The paper includes entity masking as a variable and documents inter-annotator agreement scores, which provides useful context. However, the modest agreement score (averaging 0.55) raises questions about the reliability of the ground truth data against which LLM performance is measured.

A methodological gap is the lack of detailed information about dataset retrieval. While the authors mention sourcing paragraphs from Swissdox@LiRI, a Swiss media corpus, they provide insufficient detail about search criteria, selection parameters, or sampling strategy. Could the authors add this information, at least, in the Appendix?

Other Critical Points

The paper identifies performance variations across different statements but offers limited analysis of these differences. The consistent finding that stances toward air travel reduction were better detected merits deeper exploration of potential causal factors.

I think that the paper lack of qualitative error analysis. It would benefit from a more systematic examination of where and why LLMs fail at stance detection. Such analysis could reveal patterns in misclassifications, identify challenging linguistic features, and provide practical guidance for researchers applying similar methods to other contexts. Also a brief analysis of differences between LLM and human stance detection remain underexplored.

I found also confusing the absence of a dedicated limitations section. While limitations are mentioned throughout the manuscript, this lack makes it difficult for readers to fully appreciate the constraints of the study, considering the methodological nature of the study. A consolidated discussion of methodological, conceptual, and practical limitations would enhance the paper's scholarly rigor and utility for other researchers.

Broader Implications

The research has implications for social science methodology, suggesting that researchers with limited computational resources might leverage smaller models with well-designed prompting strategies. However, the performance levels achieved (F1 scores up to 0.73 in the best case) indicate substantial room for improvement before such methods could reliably replace human coding.

The authors acknowledge the reductive nature of mapping complex political positions to discrete categories. This tension, inherent in computational approaches to discourse analysis, deserves more critical examination regarding the validity of LLM-based stance detection for drawing substantive political science conclusions.

Furthermore, the paper could benefit from a more explicit discussion of how the findings contribute to the broader field of political discourse analysis beyond the technical achievement. The authors could more thoroughly connect their methodological insights to substantive questions in political communication, ideology formation, or public opinion research.

Also a discussion of ethical implications is missing. The authors briefly mentioned to use opensource LLMs due to ethical concerns and acknowledge bias concerns through their entity masking experiments. In general, given the political nature of stance detection, the manuscript would be strengthened by a more robust discussion of potential biases and ethical implications. The use of LLMs to classify political positions raises important questions about algorithmic fairness, the potential amplification of existing biases in political discourse, and the ethics of automated categorization of political actors' stances. A deeper engagement with these ethical dimensions would enhance the paper's contribution.

Overall Assessment

In general, I found this manuscript very fascinating and helpful for social scientists who want to introduce LLMs in their research, in particular for content annotation tasks.

**Reviewer #2: **The authors set out to explore zero‐shot stance detection on 1,710 German newspaper paragraphs about sustainable urban transport, comparing seven prompt‐chain strategies across four variants of Llama3 models. They report a top weighted‐macro F1 of 0.73 and highlight the importance of prompt design, model size, and statement‐specific performance differences.

However, the manuscript offers little in the way of novel methodological contribution. Many of the prompt‐chain ideas (e.g., summary‐then‐classify, hierarchical checks) closely mirror existing “chain‐of‐thought” or multi‐stage prompting techniques already explored in recent LLM literature [16,17,26], yet the authors neither clearly delineate how their chains differ nor benchmark against those prior works.

Without a compelling new algorithmic insight or a head‐to‐head comparison with fine‐tuned or few‐shot baselines, the paper reads as an incremental application rather than a substantive advance.

The annotation methodology also raises serious questions. Reported inter‐annotator agreements ranged from 0.41 to 0.63 (mean 0.55), underscoring the inherent ambiguity of the task even for humans. Yet the authors do not analyze how this noise propagates into model evaluation, nor do they provide a human‐performance benchmark for comparison. Critically, they exclude their initial batch used to build the codebook, which may bias the remaining “gold” data toward easier cases. Furthermore, the annotation codebook—while included—lacks sufficient detail on borderline examples, making replication and assessment of labeling quality impossible.

Evaluation is similarly limited. The study is confined to a single domain (sustainable transport) and one language (German), with no experiments to test generalizability across topics, styles, or languages. There is no statistical testing of differences between prompt chains or models, and no thorough error analysis to reveal systematic failures (e.g., on sarcasm or implicit statements). The absence of any computational‐cost or latency measurements further hinders assessment of real‐world feasibility.

Finally, the manuscript’s presentation suffers from overlong, jargon‐laden passages and insufficient clarity on critical details: exact prompt templates are buried in the appendix; hyperparameter settings, inference details, and code‐release links are not readily accessible; and key tables (e.g., per‐class metrics) are omitted. Also, the figures are missing.

In summary, while the topic of applying LLMs to stance detection is timely, the paper in its present form falls short on novelty, methodological rigor, evaluation details, and reproducibility. I therefore recommend rejection.

**Reviewer #3:** Summary

This paper explores zero-shot stance detection using German LLaMA3-based LLMs applied to political discourse in German newspaper texts. It proposes several prompt chain strategies and evaluates them across five policy-related statements, offering insights into model performance variation and the utility of summarization and entity masking.

Strengths

1. Timely and relevant topic bridging NLP and political discourse analysis.

2. Use of real-world, manually annotated German-language data.

3. Systematic evaluation of multiple prompt chain strategies.

4. Open-source code and data enhance reproducibility.

Weaknesses

1. Limited Language Scope: The study only uses German data and models; cross-lingual validation or multilingual experiments would improve generalizability.

2. Unclear Prompt Design Rationale: Prompt chains are heuristically designed without sufficient justification or ablation.

3. Missing Bias Analysis: No examination of potential LLM bias, despite using politically sensitive content.

4. Low Annotation Agreement: Inter-annotator agreement is modest (mean IAA = 0.55), raising questions about label clarity and task definition.

5. Overstated Conclusions: Claims about general applicability are not fully supported given the narrow empirical setup.

6. Figures Lack Clarity and Professionalism: Several figures (e.g., Fig. 11–13) are cluttered, use inconsistent formatting, and do not follow standard academic design conventions, making them hard to interpret.

Overall, the paper addresses an important problem and has potential. However, a major revision is needed to improve comparative evaluation, clarify methodology, and broaden the empirical scope.

6. PLOS authors have the option to publish the peer review history of their article (what does this mean?). If published, this will include your full peer review and any attached files.

Reviewer #1: No

Reviewer #2: No

Reviewer #3: No

---

## [Author Response · Author response to Decision Letter 1]

2 Oct 2025

All responses to specific comments are available in the required, attached response letter.

---

## [Editor Report · Decision Letter 1]

14 Oct 2025

Promises and pitfalls of using LLMs to identify actor stances in political discourse

PONE-D-25-07214R1

Dear Dr. Angst,

We’re pleased to inform you that your manuscript has been judged scientifically suitable for publication and will be formally accepted for publication once it meets all outstanding technical requirements.

Kind regards,

Thomas W. Sanchez, PhD

Academic Editor

PLOS ONE
---

## [Editor Report · Acceptance letter]

PONE-D-25-07214R1

PLOS ONE

Dear Dr. Angst,

I'm pleased to inform you that your manuscript has been deemed suitable for publication in PLOS ONE. Congratulations! Your manuscript is now being handed over to our production team.

Kind regards,

on behalf of

Dr. Thomas W. Sanchez

Academic Editor

PLOS ONE